# Effect of Potassium Doping on the Structural and Catalytic Properties of Co/MnO_x_ Catalyst in the Steam Reforming of Ethanol

**DOI:** 10.3390/ma16155377

**Published:** 2023-07-31

**Authors:** Magdalena Greluk, Marek Rotko, Grzegorz Słowik, Sylwia Turczyniak-Surdacka, Gabriela Grzybek, Katarzyna Tyszczuk-Rotko

**Affiliations:** 1Department of Chemical Technology, Faculty of Chemistry, Maria Curie-Sklodowska University in Lublin, Maria Curie-Sklodowska Sq. 3, 20-031 Lublin, Poland; 2Biological and Chemical Research Centre, University of Warsaw, 101 Żwirki i Wigury Street, 20-089 Warsaw, Poland; 3Faculty of Chemistry, Jagiellonian University, Gronostajowa 2, 30-387 Krakow, Poland; 4Department of Analytical Chemistry, Faculty of Chemistry, Maria Curie-Sklodowska University in Lublin, Maria Curie-Sklodowska Sq. 3, 20-031 Lublin, Poland

**Keywords:** ethanol steam reforming, hydrogen production, cobalt-based catalysts, manganese oxides, potassium promoter

## Abstract

The promotional effect of potassium (~1.25 wt%) on a Co/MnO_x_ catalyst was studied for samples prepared by the impregnation method in the steam reforming of ethanol (SRE) process at 420 °C for a H_2_O/EtOH molar ratio of 12/1. The catalysts were characterized using physicochemical methods to study their textural, structural, and redox properties. The XRD studies revealed that, during the treatment of both cobalt-based catalysts under a hydrogen atmosphere at 500 °C, Co^0^ and MnO phases were formed by the reduction in Co_3_O_4_ and Mn_2_O_3_/Mn_3_O_4_ phases, respectively. Potassium doping significantly improved stability and ability for the C–C bond cleavage of the Co/MnO_x_ catalyst. The enhancement of activity (at ~25%) and selectivity to hydrogen (at ca. 10%) and the C1 product, mainly carbon dioxide (at ~20%), of the Co/MnO_x_ catalyst upon potassium doping was clarified by the alkali promoter’s impact on the reducibility of the cobalt and manganese oxides. The microscopic observations revealed that fibrous carbon deposits are present on the surface of Co/MnOx and KCo/MnOx catalysts after the SRE reaction and their formation is the main reason these catalysts deactivate under SRE conditions. However, carbon accumulation on the surface of the potassium-promoted catalyst was ca. 12% lower after 18 h of SRE reaction compared to the unpromoted sample.

## 1. Introduction

Hydrogen is considered an ideal energy carrier in fuel cell technology because of its high efficiency. Hydrogen is mainly produced from the steam reforming of fossil fuels, which emit harmful air pollutants into the atmosphere. The steam reforming of ethanol (SRE), C_2_H_5_OH + 3H_2_O → 6H_2_ + 2CO_2_, is of interest since it provides a promising method of hydrogen production from renewable resources such as bioethanol, which are easily obtained from biomass. The hydrogen production efficiency first depends on the nature of the catalyst used in the SRE reaction. The preferred catalysts should not be poisoned by carbon monoxide at the reaction temperature, should allow for a fast oxygen transfer and the formation of oxygen vacancy defects, and should contain metals that remain highly active after numerous oxidation-reduction cycles. Cobalt-based catalysts are proposed as more cost-efficient substitutes with a similar C–C bond scission ability in the SRE process to noble-metal-based catalysts. However, their drawback is significant deactivation, which occurs under SRE conditions mainly due to oxidation and sintering of the metallic active phase, the migration of cobalt ions into the support to form nonreducible mixed-metal cobalt oxides such as aluminates or silicates, and the accumulation of surface carbon species that poison the catalyst by blocking the active surface cobalt sites.

The support materials may strongly influence the catalytic performance of cobalt-based catalysts in the SRE reaction when allowing for the suppression of both cobalt sintering and coke formation [1,2,3,4,5,6]. Moreover, reducible oxides such as ZnO and CeO_2_ provide a spillover of oxygen species to the metal particles dispersed on their surfaces, thereby facilitating the oxidation of the carbon deposit [2,3,5,7,8,9,10,11]. However, although manganese oxides possess plenty of active lattice oxygen species that play a role in the efficient elimination of carbon deposition, there are not many reports [1,12,13,14,15,16] on their use as a support for the catalysts of SRE reaction. Both types of manganese oxides with layered and tunnel structures were the subject of this research [12,14,15]. Conventional methods were used to prepare these materials, such as redox precipitation to obtain tunneled structured potassium containing cryptomelane-type manganese dioxide [15], comproportionation reactions of Mn^2+^ and MnO_4_^−^ under alkaline conditions to obtain layer-structure birnessite-type manganese oxide [12,14], or the hydrothermal treatment of layered manganese oxide precursor (birnessite) to obtain tunneled-structure todorokite-type manganese oxide [14,16]. Unfortunately, all these preparative processes of layer- or tunnel-structure manganese oxide have multiple steps and/or are time-consuming. Moreover, all these complex layer- and tunnel-structures of manganese oxide, during hydrogen pretreatment at elevated temperatures, are reduced to lower valence states, an effect that generally gives rise to strong structural and surface changes. The decomposition of the manganese oxides phase into a stable lower valent during the reduction can lead to the collapse of the tunnel structure. For example, Gac et al. [15] indicated that a reduction in the cryptomelane-type manganese oxide led to the development of large, cube-like MnO particles and the disappearance of its tunnel structure. As the activation of the SRE catalysts with hydrogen at elevated temperatures leads to a reduction in the manganese oxides to lower oxidation states with the concomitant evolution of oxygen and complete structural rearrangement, Greluk et al. [16] proposed a simple precipitation method for the preparation of a mixture of chemically simple and inexpensive manganese oxides, Mn_2_O_3_ and Mn_3_O_4_, as a support for the SRE catalyst.

A limitation of the coke deposition might be also achieved by the addition of alkali promoters [16,17,18,19], which can significantly change both the structure and surface of the catalyst. The key role of alkali metal additives is dependent on the nature of the catalytic material, e.g., the type and crystallite size of the active phase, the phase composition and the morphology of the support. Moreover, because of the high reactivity and volatility of alkalis, the high-temperature pretreatment of SRE catalysts and the interaction of numerous reagents of the SRE process can result in undesired processes such as alkali surface redistribution, a solid-state reaction with the active phase/support, their agglomeration, or even alkali promoters’ desorption, affecting the resulting performance of the catalyst. Therefore, the promotional effect of alkali must be elucidated individually for each catalytic system [17,20]. 

To our knowledge, no studies were conducted to investigate the impact of the potassium promoter on manganese-oxide-supported, cobalt-based catalysts in the SRE reaction. Therefore, this work aimed to determine the role that potassium promotion plays in the performance of the Co/MnO_x_ catalyst during the SRE process. The chemical and physical properties of catalysts were characterized by BET, XRF, XRD, TPR, SEM, TEM and XPS methods.

## 2. Materials and Methods

### 2.1. Materials

The materials used in the preparation of samples: manganese(II) acetate tetrahydrate (≥97.0%, Sigma-Aldrich, Darmstadt, Germany), cobalt(II) nitrate hexahydrate (≥98.0%, Sigma-Aldrich), potassium nitrate (≥99.0%, Merck, Darmstadt, Germany) and ammonium carbonate (100%, Avantor, Gliwice, Poland) were used without further purification.

### 2.2. Preparation of Catalysts

The MnO_x_ support was prepared using the precipitation method with 1 mol/L ammonium carbonate solution as a precipitant from the precursor of manganese acetate. Ammonium carbonate solution was added dropwise to manganese acetate solution with vigorous stirring until a pH of 8 was maintained at 40 °C. The resulting precipitate was aged at 60 °C for 2 h. The suspension was filtered and then washed several times with deionized water until reaching a pH of 7. The precipitate was then dried at 110 °C and calcined at 500 °C in air.

The Co/MnO_x_ catalysts were prepared using the impregnation method by mixing cobalt nitrate solution and MnO_x_ support to achieve 10 wt% cobalt loading. The sample was dried at 110 °C and calcined at 500 °C in air.

The KCo/MnO_x_ catalysts were prepared using the impregnation method by mixing potassium nitrate solution and Co/MnO_x_ sample to achieve 2 wt% potassium loading. The sample was dried at 110 °C and calcined at 500 °C in air.

A schematic drawing of the preparations of MnO_x_ support and Co/MnO_x_ and KCo/MnOx catalysts is presented in Figure 1.

### 2.3. Catalysts Characterization

Details about the physicochemical characterization of materials can be found in our previous work [16]. The metal loading of catalysts was determined by the X-ray fluorescence method, using an Axios mAX (PANalytical, Malvern, UK) fluorescence spectrometer. The BET surface area, pore volume, and pore diameter of the samples were measured by ASAP 2405N (Micromeritics, Norcross, GA, USA) instrument, using nitrogen adsorption/desorption isotherms collected at −196 °C. The X-ray powder diffraction (XRD) patterns were obtained with an Empyrean X-ray (PANalytical, Malvern, UK) diffractometer. Temperature-programmed reduction (TPR) studies were carried out with an AutoChem II 2920 (Micromeritics, Norcross, GA, USA) analyzer. The high-resolution transmission electron microscopy (HRTEM) images of catalysts were obtained with an electron transmission microscope Titan G2 60–300 kV (FEI Company, Hillsboro, OR, USA). X-ray photoelectron spectroscopy (XPS) studies were performed in a Kratos Axis Supra Spectrometer (Kratos Analytical, Manchester, UK) equipped with a monochromatized Al source. 

### 2.4. SRE Experimental Tests

The catalyst activity test was carried out in a quartz tube reactor (inner diameter: 10 mm) using a Microactivity Reference unit (PID Eng & Tech., Alcobendas, Spain). The reactor was filled with a catalyst sample diluted with quartz. A total of 100 mg of catalyst sample was reduced with hydrogen at 500 °C for 1 h. After this reduction, the performance was evaluated at atmospheric pressure, a temperature of 420 °C and a H_2_O/EtOH molar ratio of 12/1, with a total flow rate equal to 100 mL min^−1^. The reaction substrates and products were analyzed by online gas chromatographs (Bruker 430-GC and Bruker 450-GC, (Bruker, Billerica, MA, USA) equipped with thermal conductivity detectors (TCD).

The conversion of ethanol, conversions of ethanol into individual carbon-containing products and the selectivity of hydrogen formation were calculated based on the previously reported equations [16]. 

## 3. Results and Discussion

### 3.1. The Effect of Potassium Doping on the Catalysts’ Physicochemical Properties

The results of BET analysis, quantitative analysis (analyzed by XRF method), and values of average crystallite size (analyzed by XRD and TEM methods) of Co/MnO_x_ and KCo/MnO_x_ catalysts are summarized in Table 1. The desired loading of cobalt (10 wt%) was obtained with good accuracy. However, the potassium promoter content is less than the intended loading of 2 wt%. The Co/MnO_x_ catalyst exhibits a low surface area, which decreases by more than 50% following re-calcination after the addition of a potassium promoter. Because the low surface area of support does not facilitate the dispersion of metal particles, both Co/MnO_x_ and KCo/MnO_x_ catalysts exhibit a large Co^0^ particle size (see also Appendix A, Appendix A). The slightly larger size of Co^0^ particles for KCo/MnO_x_ catalysts results from the agglomeration of the initially formed cobalt species crystallites of the Co/MnOx catalyst during its second calcination, after impregnation with potassium salt [18]. STEM-EDS maps of samples, after their reduction with hydrogen at 500° (Figure 2 and Figure 3), reveal a comparatively homogenous cobalt dispersion on the MnO_x_ support of both catalysts, as well as potassium accumulation on both MnO_x_ and Co^0^ particles for the K-doped sample.

Figure 4A,B shows the XRD patterns of the calcined Co/MnO_x_ and KCo/MnO_x_ catalysts, which reveal that both Co/MnO_x_ and KCo/MnO_x_ samples contain a mixture of tetragonal Mn_3_O_4_ (JCPDS file no. 04-005-9818) and cubic Mn_2_O_3_ phases (JCPDS file no. 00-001-1127). However, in the case of both catalysts, the major species is Mn_2_O_3_. Diffraction peaks observed at diffraction angles (2θ) of 24.0°, 33.1°, 38.6°, 45.3°, 49.5°, 55.2°, 60.9° and 66.0° can be assigned to the (211), (222), (400), (332), (431), (400), (611) and (622) crystalline planes of the cubic Mn_2_O_3_ crystal. Similarly, prominent diffraction peaks appeared at 2θ of 28.0°, 31.4°, 32.2°, 36.1°, 44.8°, 58.8°, 60.0°, which can be assigned to the (112), (200), (103), (211), (220), (321) and (224) crystalline planes of the tetragonal Mn_3_O_4_ phase. Moreover, because the main reflection for Mn_3_O_4_ at 2θ = 33.1° is nearly identical to the main reflection of Co_3_O_4_ (JCPDS file no. 04-002-0644) at 2θ = 36.1°, the possibility that these both reflections overlap should not be excluded. However, the XRD pattern of both catalysts mainly diffracts at angles that can attributed to the phases of support due to its much higher content compared to the Co_3_O_4_ active phase. Upon the reduction in both cobalt-based catalysts for 2 h at 500 °C, the XRD peaks corresponding to the cubic-phase Mn_2_O_3_ and tetragonal-phase Mn_3_O_4_ disappeared completely, and new reflections that appeared at 34.9°, 40.4°, 58.5°, 70.0 and 73.6° are mainly present, which can be assigned to the (111), (200), (220), (311) and (222) planes of the cubic structure of MnO (JCPDS file no. 00-075-1090) (Figure 4C,D). In addition to the predominant diffraction peaks derived from the MnO support, weaker diffraction lines at 2θ = 43.7° and 51.2°, corresponding to the reflection planes of the Co^0^ phase, are visible [16,21,22,23,24,25,26,27,28]. Moreover, the absence of any potassium phase for the KCo/MnO_x_ catalyst in the XRD patterns may result from several factors, namely its considerably low amount and/or amorphous form and/or high dispersion over the catalyst surface [16,29]. Unfortunately, due to the low potassium-promoter content, there is a problem with the understanding of its role, since it is difficult to characterize this on the catalytic surface [29]. 

Detailed HRTEM measurements (Figure 5 and Figure 6) were taken for both cobalt-based catalysts after their reduction at 500 °C, and the results are in good agreement with the XRD findings. Well-crystallized Co^0^ and MnO phases are identified for both Co/MnO_x_ (Figure 5) and KCo/MnO_x_ (Figure 6) catalysts. For both catalysts, HRTEM images reveal lattice fringes with an interplanar spacing of 0.205, 0.222 and 0.225 nm, which may be ascribed to the (111) plane of Co^0^, (200) and (111) planes of MnO, respectively. Moreover, for the Co/MnO_x_ sample, lattice fringes with an interplanar spacing of 0.157 nm, which can be ascribed to the (220) planes of MnO, are also observed. The planes of both samples were also identified by the spots on the corresponding FFT pattern.

The reducibility of the MnO_x_ support, Co/MnO_x_ and KCo/MnO_x_ catalysts, was also studied by the H_2_-TPR method, and the results are displayed in Figure 7. According to the literature [16,30,31,32,33], bare MnO_x_ exhibits a low-temperature peak at 220–330 °C, which can be assigned to the reduction from Mn_2_O_3_ to Mn_3_O_4_ and high temperature at ca. 330–520 °C, corresponding to the transformation of Mn_3_O_4_ to MnO. Based on the study of the literature [34,35], due to two reduction steps, the H_2_-TPR profile of Co_3_O_4_ has a low-temperature peak below 300 °C, assigned to the reduction from Co_3_O_4_ to CoO, and a high-temperature peak between 300 and 700 °C, corresponding to the further reduction from CoO to Co^0^. This means that the reduction peaks in manganese oxides overlapped with those of cobalt oxides and could not be distinguished. The H_2_-TPR profile of the Co/MnO_x_ catalyst has two regions of hydrogen consumption: a low-temperature region at 220–310 °C and a high-temperature region at 310–750 °C, which could correspond to the reduction from Mn_2_O_3_ to Mn_3_O_4_ and Co_3_O_4_ to CoO, and the reduction from Mn_3_O_4_ to MnO and CoO to Co^0^, respectively. Moreover, the broad peak observed in the high-temperature region (above 500 °C) could be a result of the synergetic interactions between Mn and Co species [36]. A reduction in the K-containing sample leads to a lower temperature compared to a reduction in Co/MnO_x_ material. In addition, two reduction peaks were observed that could be ascribed to the stepwise reduction in manganese and cobalt oxides. Moreover, a small shoulder below 260 °C is related to the presence of surface-active species with different Mn–O bond strengths [16,37]. The KCo/MnOx sample (Table 1) exhibits high hydrogen consumption, which may result from the reduction in nitrates stored on the catalyst’s surface, facilitated by the presence of potassium [16]. It can be supposed that the nitrate reducibility is increased by the presence of cobalt and hydrogen, which is dissociated upon the Co^0^ particles’ spilling onto the nitrate, reducing their quantity [38].

### 3.2. The Effect of Potassium Doping on the Performance of the Catalysts in the Ethanol Steam Reforming Reaction

The effect of potassium doping on the cobalt-based catalyst in the SRE process was determined by ethanol conversion (Figure 8) and selectivity to products (Figure 9) at the temperature of 420 °C for a H_2_O/EtOH molar ratio of 12/1. The initial complete conversion of ethanol decreases after 18 h of the SRE process to ca. 40 and 70% over Co/MnO_x_ and KCo/MnO_x_ catalysts, respectively. At the beginning of the SRE reaction, the KCo/MnO_x_ catalyst exhibits high selectivity to hydrogen and carbon dioxide. In contrast, hydrogen and carbon monoxide are mainly produced in the presence of the Co/MnO_x_ sample, which suggests an increase in activity in water gas shift (WGS, Reaction (1)) reactions in the presence of a potassium-doped catalyst:CO + H_2_O ↔ CO_2_ + H_2_(1)

Furthermore, the C2 and C3 products, i.e., acetaldehyde, ethylene and acetone, are present among the by-products of a Co/MnO_x_ sample from the beginning of the SRE reaction, which indicates the poor ability of this catalyst in terms of the C–C bond cleavage. The addition of potassium allows for restrictions to the production of these products and, initially, only small amounts of acetaldehyde are formed over the KCo/MnO_x_ sample. However, the amount of this product drastically increases with the increase in time-on-stream over both cobalt-based catalysts. After 18 h of the SRE process, selectivity to acetaldehyde is ca. 70 and 55% for Co/MnO_x_ and KCo/MnO_x_ catalysts, respectively. Next to hydrogen, it is the second product that is mainly produced in the presence of these catalysts during the SRE process. This indicates that the ability of both samples in the C–C bond scission decreases with the increase in time-on-stream because of the deactivation of cobalt active sites on the catalysts’ surface under SRE conditions. However, the lower selectivity to the C2 and C3 products and higher selectivity to the C1 products suggests that more cobalt-active sites remained available on the surface of the KCo/MnO_x_ catalyst to break the C–C bond. 

### 3.3. The Effect of Potassium Doping on Prevention of the Cobalt-Based Catalyst Deactivation under SRE Conditions

The microscopic (Figure 10 and Figure 11) observations revealed that fibrous carbon deposits are present on the surface of Co/MnO_x_ and Kco/MnO_x_ catalysts. The cobalt nanocrystals are almost completely detached from the MnO_x_ support by carbon deposits, and they are either confined to the fibres or located at the tip of the carbon filaments, being totally or partially surrounded by a carbon shell.

Based on the HRTEM images and the corresponding phase identification obtained using the FFT method (Figure 11), two forms of cobalt-active phase were identified in the case of the spent Co/MnO_x_ catalyst, namely Co^0^ and CoO. However, the Co^0^ form of the cobalt-active phase was detected for the spent KCo/MnO_x_ catalyst. Moreover, support for the Co/MnO_x_ sample was identified as including two forms, MnO and Mn_2_O_3_, where only the MnO phase was detected for the support of the spent KCo/MnOx catalyst.

Figure 12 shows the amount of carbon deposits that were formed on the surface of the Co/MnO_x_ and KCo/MnO_x_ catalysts as a function of time. The rate of carbon formation significantly decreased in the presence of a potassium promoter on the surface of a cobalt-based catalyst, indicating that its addition can inhibit carbon deposition and/or promote gasification [16]. Potassium species probably migrated and covered the surface of the Co^0^ particles, which favoured the adsorption-dissociation of the H_2_O molecule, and therefore increased the direct oxidation of the carbonaceous species [39].

The survey spectra of spent cobalt-based catalysts (H_2_O/ethanol = 12/1, 420 °C) show the major signals originating from C, O and Mn elements, and minor signals originating from Co. The atomic surface coverage by carbon deposits was estimated at around 81.5 at.% for Co/MnOx and 91 at.% for KCo/MnO_x_ catalysts. Most of the carbon exists in graphitic-like compounds, as confirmed by the asymmetric C 1s peak shape (284.5 eV, not shown here). The high carbon coverage of the surface leads us to expect that most crystallites of the active phase and support were occluded in carbon; therefore, their oxidation state remained unchanged even upon exposure to air (Figure 13). The Co 2p3/2 spectrum can be resolved into components, which are attributed to the Co–Co (778.4 eV) and Co–O bonding (780.4 eV), respectively [40,41]. A higher concentration of metallic cobalt species can be observed for the potassium-promoted catalyst, suggesting that this promotor plays a very important role in the reduction in the active phase. For the manganese oxidation state, the literature data [41] suggest that MnO and Mn_2_O_3_ phases are present. The relative concentration of Mn(II)/(Mn(II) + Mn(III)) in both samples is very similar, in the range of 51%. The interpretation of the results obtained for bare manganese support requires a comparison of both the spectrum recorded for the as-calcined sample and after ESR reaction. This is important, as bare support was rather inactive in the ESR [18], whereas the survey spectrum suggests almost 55 at.% coverage of the surface by carbon-containing species. This result seems less surprising when one notes that the atomic contribution of adventitious carbon species on the surface of the calcined sample is equal to 39 at.%. The calculation of the relative concentration of Mn(II) in the overall Mn 2p3/2 spectrum suggests a lower concentration of these species (43%) as compared to the results obtained for cobalt-based catalysts. 

## 4. Conclusions

The doping effect of a potassium-promoter on the novel Co/MnO_x_ catalytic material’s performance in the SRE reaction was studied, focusing on the catalyst’s ability to resist carbon growth. It is beneficial to consider the significant increase in catalytic stability achieved by the Kco/MnO_x_ catalyst in comparison with the Co/MnO_x_ sample, and the distribution of the products over these catalysts under SRE conditions. Compared with the unpromoted Co/MnO_x_ catalyst, the Kco/MnO_x_ sample indicated an increase in the selectivity to two main products of the SRE reaction, i.e., hydrogen (Co/MnO_x_—ca. 55% and Kco/MnO_x_—ca. 70%) and carbon dioxide (Co/MnO_x_—ca. 20% and Kco/MnO_x_—ca. 40%). In turn, selectivity to C2 products was ca. 20% higher in the presence of an unpromoted Co/MnO_x_ sample, indicating that more cobalt-active sites remained available on the surface of the KCo/MnO_x_ catalyst to cleave the C–C bond. Although fibrous carbon was present on the surface of both Co/MnO_x_ and KCo/MnO_x_ samples, the carbon accumulation was ca. 12% higher on the surface of unpromoted catalysts after 18 h of SRE reaction. This indicates that potassium species probably migrated and covered the surface of the Co^0^ particles, which favoured the adsorption–dissociation of the H_2_O molecule, and therefore increased the direct oxidation of the carbonaceous species. It could be concluded that the crucial effect of potassium dopant is the decrease in carbon accumulation on the catalyst surface and the increase in the number of sites for water adsorption, which improve the catalyst stability. Furthermore, the alkali metal could change the electronic properties of the cobalt species, and hence facilitate the redox ability of active sites both during activation with hydrogen and under SRE conditions. The maintenance of a reduced oxidation state for the catalyst surface during the SRE process is important to ensure an effective C—C bond; thus, the catalytic stability of a potassium cobalt-based catalyst was improved. According to the results of the characterization, the addition of the alkali metal did not influence the cobalt active-phase dispersion, but was not a key parameter improving Co/MnO_x_ catalysts’ activity, selectivity and resistance to deactivation.

## Figures and Tables

**Figure 1 materials-16-05377-f001:**
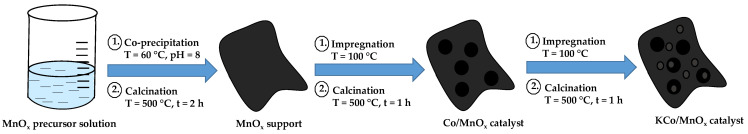
A schematic drawing of the preparations of MnO_x_ support and Co/MnO_x_ and KCo/MnO_x_ catalysts.

**Figure 2 materials-16-05377-f002:**
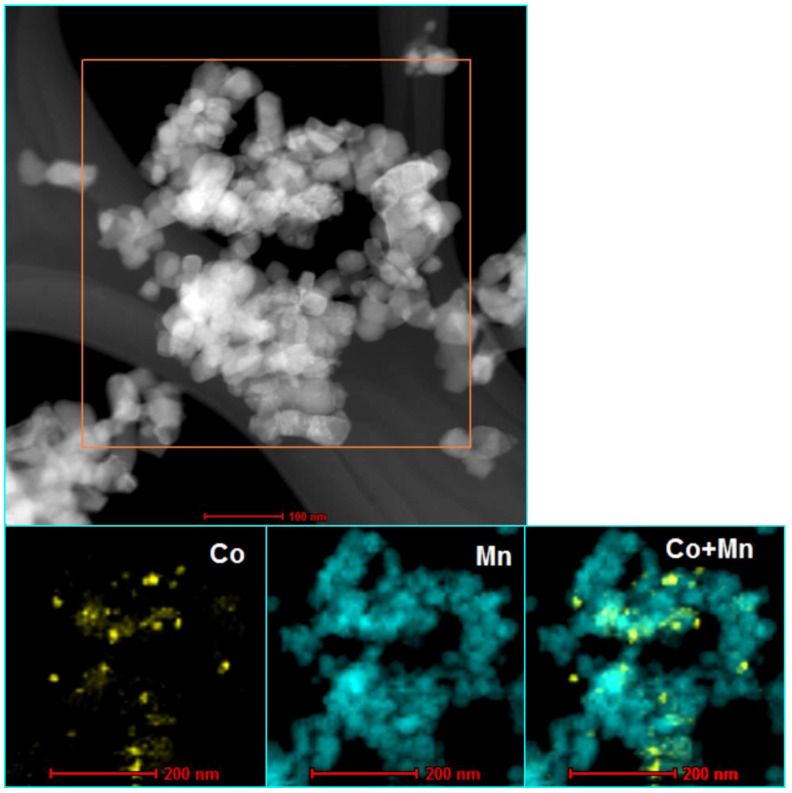
STEM-EDS analysis of Co/MnO_x_ catalyst after their reduction with hydrogen at 500 °C.

**Figure 3 materials-16-05377-f003:**
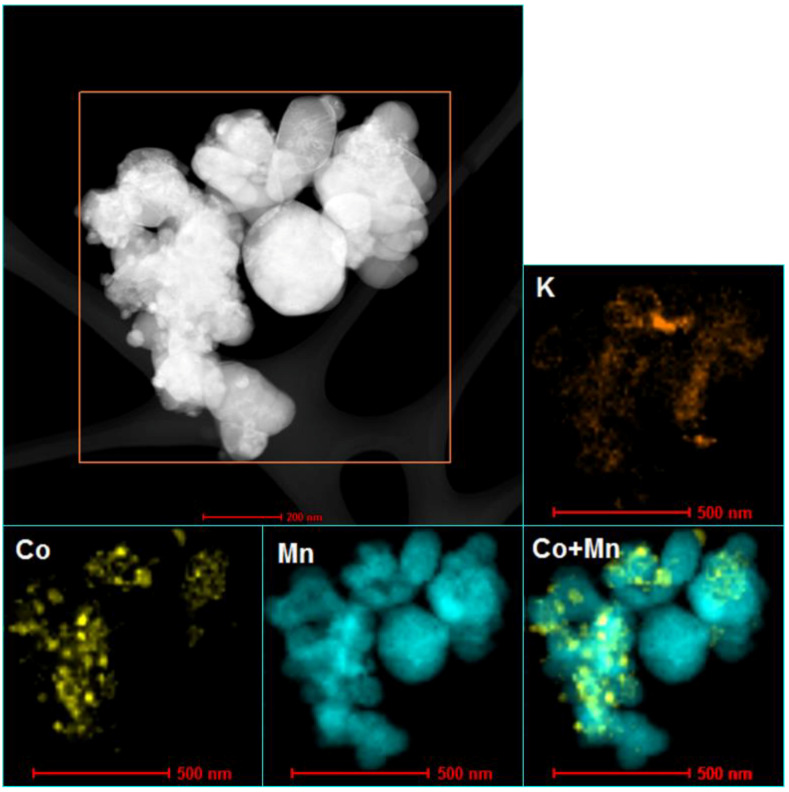
STEM-EDS analysis of KCo/MnO_x_ catalyst after their reduction with hydrogen at 500 °C.

**Figure 4 materials-16-05377-f004:**
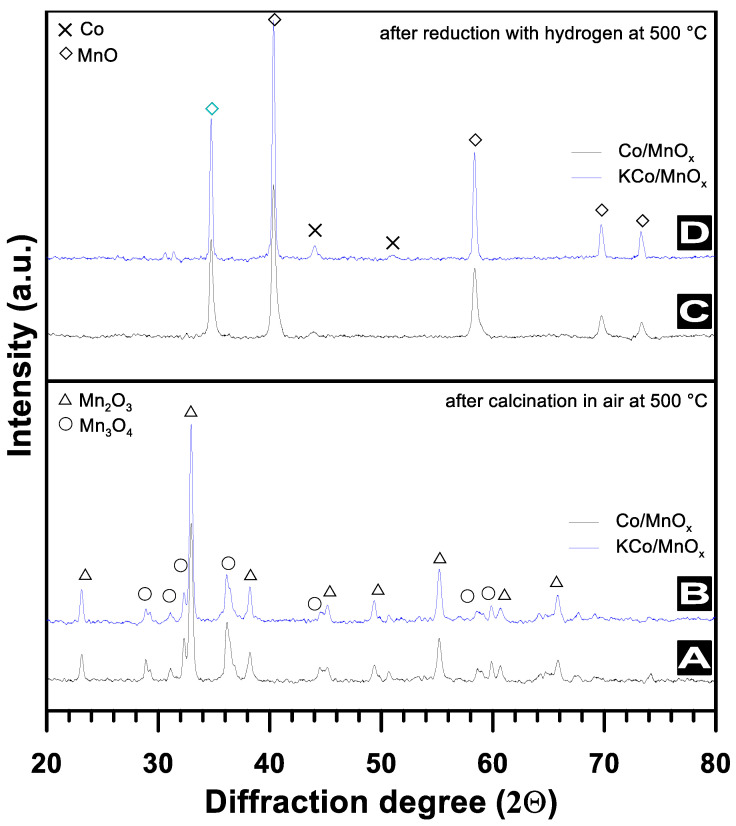
XRD patterns of Co/MnO_x_ and KCo/MnO_x_ catalysts after their calcination in-air at 500 °C (**A**,**B**) and after their reduction with hydrogen at 500 °C (**C**,**D**).

**Figure 5 materials-16-05377-f005:**
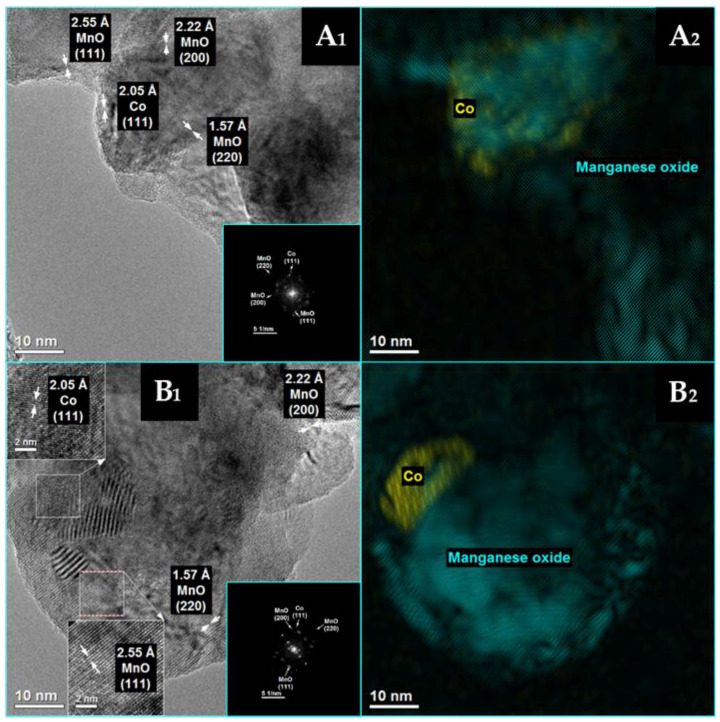
HRTEM images from different points (**A_1_**,**B_1_**) of Co/MnO_x_ catalyst after reduction with hydrogen at 500 °C with the complementary phase identification ((**A_2_**,**B_2_**), Co^0^—yellow; MnO—turquoise) and the corresponding fast Fourier transform (FFT) patterns in the turquoise squares as the inset of (**A_1_**,**B_1_**). Arrows represent distance between the two the two lattice planes. Magnified view of the selected areas was enclosed in the dashed square to show diffraction fringes and their respective interplanar distances.

**Figure 6 materials-16-05377-f006:**
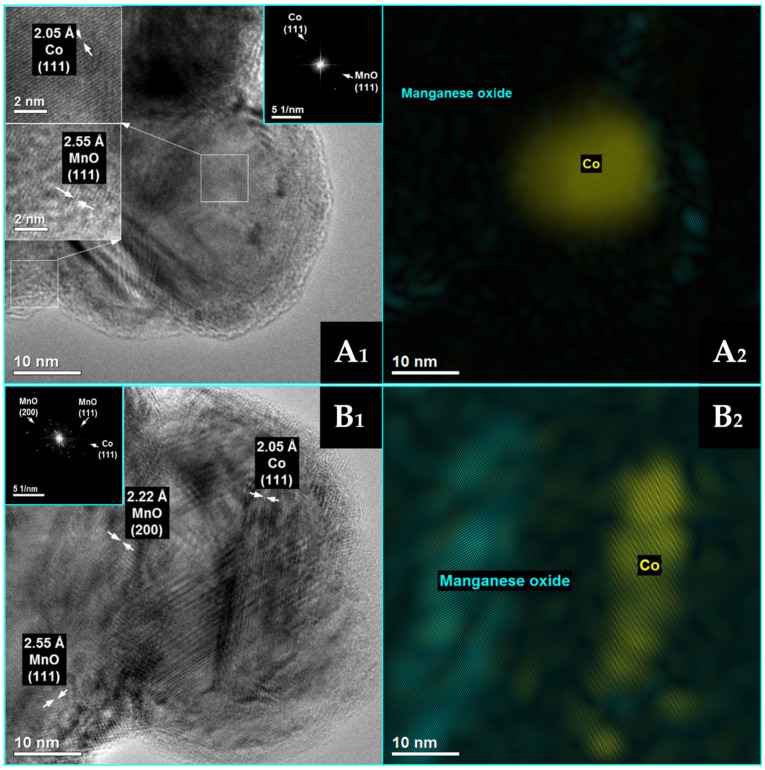
HRTEM images from different points (**A_1_**,**B_1_**) of KCo/MnO_x_ catalyst after its reduction with hydrogen at 500 °C with the complementary phase identification ((**A_2_**,**B_2_**), Co^0^—yellow; MnO—turquoise) and the corresponding fast Fourier transform (FFT) patterns in the turquoise squares as the inset of (**A_1_**,**B_1_**). Arrows represent distance between the two the two lattice planes. Magnified view of the selected areas was enclosed in the dashed square to show diffraction fringes and their respective interplanar distances. Obtained from the HRTEM image in (**A**).

**Figure 7 materials-16-05377-f007:**
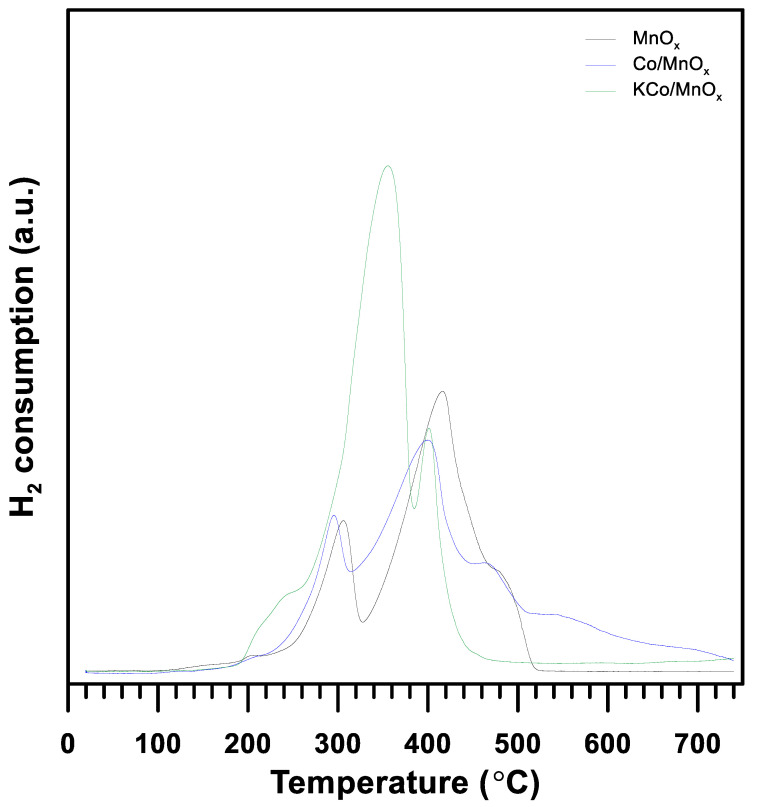
H_2_-TPR profiles of MnO_x_ support and Co/MnO_x_ and KCo/MnO_x_ catalysts.

**Figure 8 materials-16-05377-f008:**
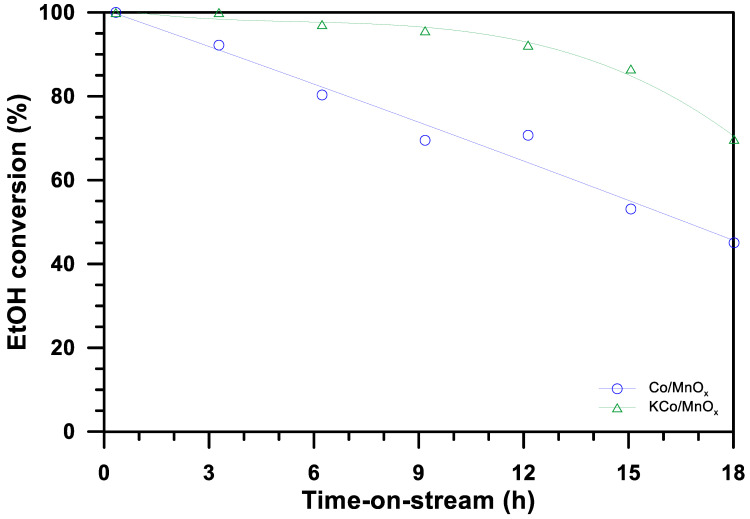
The ethanol conversion over Co/MnO_x_ and KCo/MnO_x_ catalysts after 18 h of the SRE process at 420 °C for an H_2_O/EtOH molar ratio of 12/1.

**Figure 9 materials-16-05377-f009:**
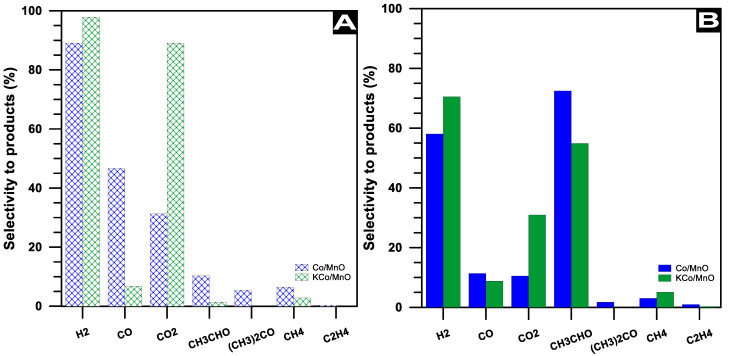
The selectivity to products at the beginning (**A**) and after 18 h (**B**) of the SRE process at 420 °C for a H_2_O/EtOH molar ratio of 12/1 over Co/MnO_x_ and Kco/MnO_x_ catalysts.

**Figure 10 materials-16-05377-f010:**
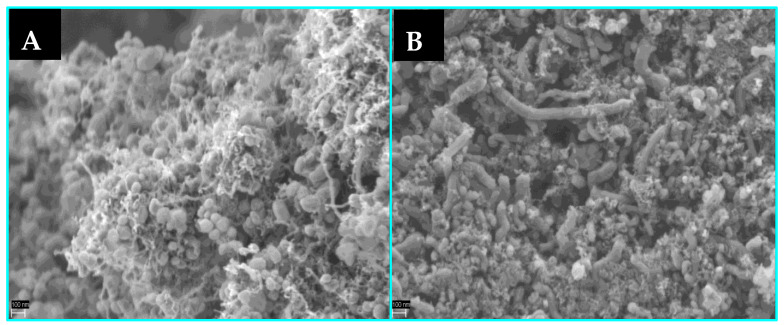
SEM images of Co/MnO_x_ (**A**) and KCo/MnO_x_ (**B**) catalyst after 18 h of SRE reaction at 420 °C for H_2_O/EtOH molar ratio of 12/1.

**Figure 11 materials-16-05377-f011:**
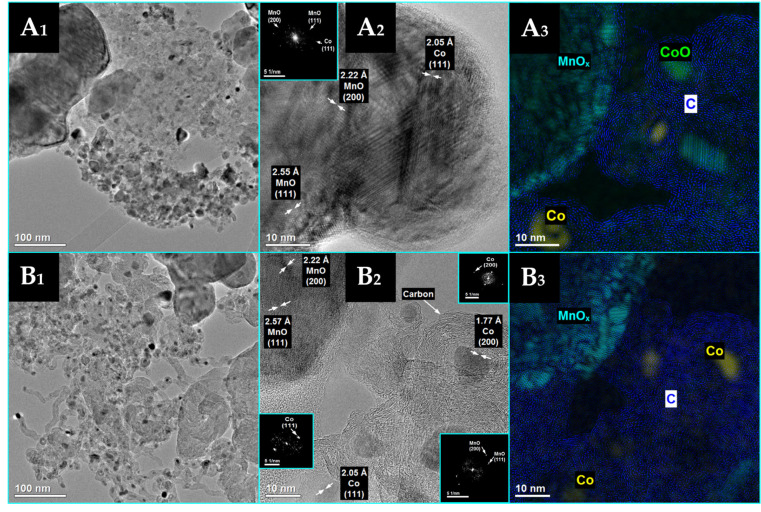
Microscopic analysis of Co/MnO_x_ (**A**) and KCo/MnO_x_ (**B**) catalyst after 18 h of SRE reaction at 420 °C for H_2_O/EtOH molar ratio of 12/1. TEM images (**A_1_**,**B_1_**), HRTEM images (**A_2_**,**B_2_**) and the corresponding phase identification (**A_3_**,**B_3_**), Co^0^—yellow; CoO—green, MnO—turquoise, C—blue).

**Figure 12 materials-16-05377-f012:**
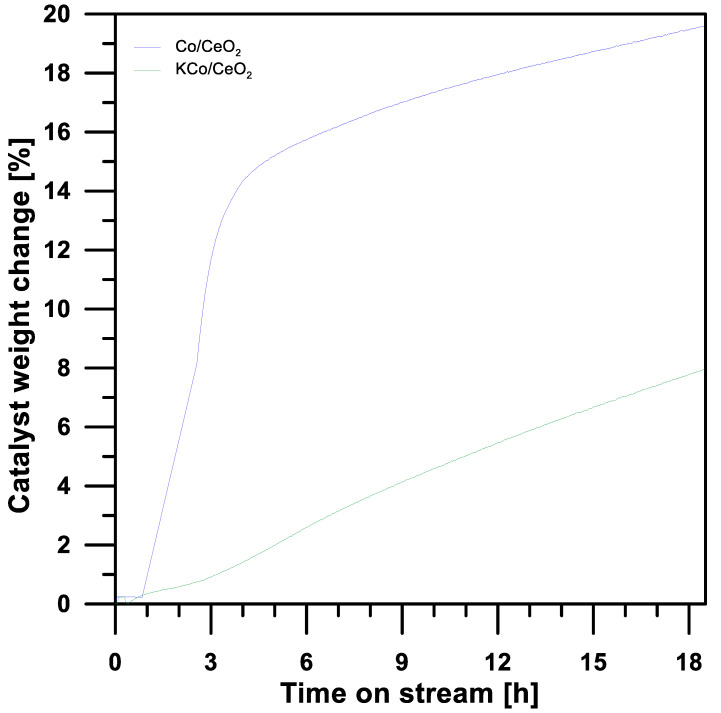
Changes in Co/MnOx and KCo/MnOx catalysts’ weight during 18 h of SRE reaction at 420 °C for H_2_O/EtOH molar ratio of 12/1.

**Figure 13 materials-16-05377-f013:**
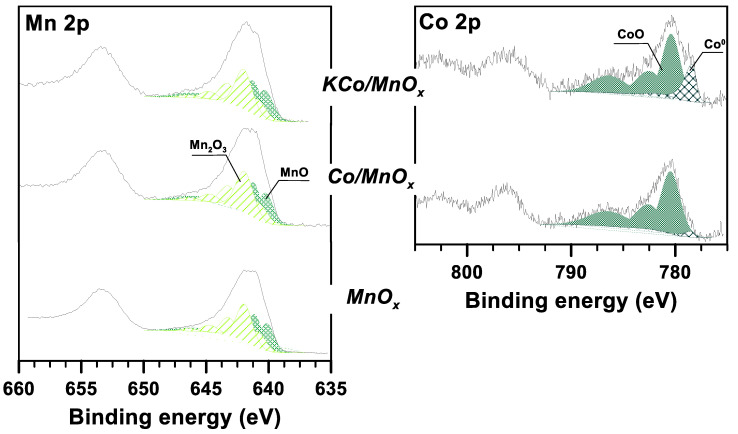
High-resolution XPS spectra of Mn 2p and Co 2p regions collected from the surface of Co/MnO_x_ and KCo/MnO_x_ catalysts after 18 h of SRE reaction at 420 °C for H_2_O/EtOH molar ratio of 12/1.

**Table 1 materials-16-05377-t001:** Physicochemical parameters for Co/MnO_x_ and KCo/MnO_x_ catalysts.

Parameter		Co/MnO_x_	KCo/MnO_x_
Metal content (wt%)		9.7	9.7
Potassium content (wt%)		-	1.25
BET surface area (m^2^/g)		11.2	5.8
Co^0^ crystallite size (nm) ^1^	By XRD	14	23
	By TEM	21	26
H_2_ consumption	Theoretical (mmol/g_Co_)	2.18	2.19
	Experimental (mmol/g)	5.21	7.23

^1^ Co^0^ denoted Co particle size for a catalyst reduced at 500 °C.

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
