# Peer review of "Effect of Potassium Doping on the Structural and Catalytic Properties of Co/MnOx Catalyst in the Steam Reforming of Ethanol"

_materials, 2023, doi:10.3390/ma16155377_

Round 1

Reviewer 1 Report

This manuscript reported that the promotion effect of potassium on Co/MnOx catalyst and its application over steam reforming of ethanol process. The topic is of interest for both academic and industrial sectors, and the quality of the research work could be adequate for publication in this journal. Nevertheless, authors must tackle and solve some important concerns.

Main comments and remarks:

1.     Numerous type-settings and grammatical faults are encountered all along the text, making it hard to read.

2.     Please pay attention to the figure notes. It hard to understand the descriptions of some figures, such as Figure 1, Figure 2, Figure 3

3.      However, the authors mainly reported the results of material characterization and catalysts evaluation, and did not explain the results scholarly, and thus some conclusions are not convincing.

Extensive editing of English language required

Reviewer 2 Report

Manuscript ID: materials-2133406

Title:  Effect of potassium doping on the structural and catalytic properties of Co/MnOx catalyst in the steam reforming of ethanol’

Comments:

Major revision

1.      In abstract various abbreviations were used. Use full form when write 1st time and then abbreviation be used throughout the manuscript. Like ‘Co/MnOx, H2O/EtOH & SRE reaction’ not clarified these terms.

2.      Abstract is too short, it should be consisting of characterization findings like SEM & XRD.

3.      What is the novelty of this work ?

4.      There are so many typo grammatical errors in whole manuscript, should be revised by some native speaker and formatting should be checked.

5.      Presentation of Figs not corrected all over the manuscript.

6.      In Figure 1 mentions name of sub figures like a, cb,c….. also scale is note visible should be redrawn. Collect the caption accordingly

7.      Figure 2 capyion not correct, also lable the XRD reaks. Insert JCPDS No also

8.      Figure 3 and 4 correct caption describe sub figures in caption

9.      In introduction more literature should be reviewed and some latest metallic catalyst should be discussed here to enhance the novelty of work like, Journal of Molecular Liquids 356 (2022) 119036, Coordination Chemistry Reviews 471 (2022) 214716.

10.  FTIR analysis should be carried out to check the structural changes of synthesized different catalysts KCo/MnOx, Co/MnOx.

11.  Introduction is short it should be include more literature about synthesis methods of Co/MnOx catalyst.

12.  There should be a sub heading 2.1 heading about materials name and purchasing companies in Heading 2.

13.   Graphical representation of synthesis scheme should be included in manuscript fpr clarity of different synthesis steps and conditions

14.  Conclusion of paper should be added to clear research work & your findings.

Average

should be revised carefully

Reviewer 3 Report

The authors presented a research paper on the effect of potassium doping on the structural and catalytic properties of the Co/MnOx catalytic system. This paper seems very interesting, well written, and organized. The rationale for the research is well outlined, and the results are likely to be quite attractive. The experimental data are presented and discussed in an understandable way. The analysis and conclusions are consistent and well supported by data and explanations. The experimental section and supporting documentation are described in sufficient detail. The work will be of broad interest to researchers working in the fields of materials science and preparative inorganic chemistry. Thus, this manuscript is worthy of publication. There are two rather minor comments, which are summarized below.

1) The caption to Figure 1 on page 4: it says that this figure is in two parts, a) and b), whereas it appears that the whole figure remains undivided in both parts.

2) Lines 157-158 separating pages 4 and 5: this sentence is rather difficult to understand (probably because of the XRD pattern analysis). It would be helpful to include a more detailed assessment of how this result relates to the surface chemistry features of the material if it contains potassium. In particular, assuming a strong oxidation of potassium on the surface of the material, this can lead to the formation of a potassium oxide state close to superoxide, such as K2O_(2+y), which may have a partially disordered structure with very low symmetry. In this case, the potassium contribution may not be visible in the XRD patterns measured for diffraction greater than 20 degrees.

Round 2

Reviewer 1 Report

The manuscript has been carefully revised according to the comments of the reviewers, and some suggested experiments have been added. I believe that It is acceptable now.

The manuscript has been carefully revised according to the comments of the reviewers, and some suggested experiments have been added. I believe that It is acceptable now.

Reviewer 2 Report

The author did not address the previous comments diligently. I again recommend the major revision

Thousands of the researchers have already done this type of research work therefore, the novelty of this work was missing; authors should discuss the novelty of their work in the introduction section. The author should made comparison between their compounds and already reported materials. Therefore I recommend some paper like Journal of Molecular Liquids 356 (2022) 119036, Coordination Chemistry Reviews 471 (2022) 214716, Molecular Catalysis 514 (2021) 111878 for guidance and comparison.

Graphical abstract is missing in the revised manuscript.

FTIR analysis is the basic characterization to support the successful synthesis of the samples. Unfortunately the author make excuse to perform FTIR analysis I strongly recommend to add FTIR analysis. if time issue, please make a request to the editor for time gain.

For the reader interest it’s necessary for author to make introduction more eye catching and interesting, therefore I made comment to add more literature about synthesis routes.

Still abstract and conclusion section lacks numerical values.

Lastly, the manuscript contain 44% similarity index which is not acceptable. Reduce it below 15%.

Moderate editing of English language required
